# Higher-Level Executive Functions in Healthy Elderly and Mild Cognitive Impairment: A Systematic Review

**DOI:** 10.3390/jcm11051204

**Published:** 2022-02-23

**Authors:** Ilaria Corbo, Maria Casagrande

**Affiliations:** 1Dipartimento di Psicologia, Università di Roma Sapienza, 00185 Roma, Italy; ilaria.corbo@uniroma1.it; 2Dipartimento di Psicologia Dinamica, Clinica e Salute, Università di Roma Sapienza, 00185 Roma, Italy

**Keywords:** Mild Cognitive Impairment, ageing, elderly, executive functions, higher-level executive functions, planning, reasoning, fluid intelligence, problem solving

## Abstract

Mild Cognitive Impairment (MCI) is a clinical syndrome characterized by a moderate decline in one or more cognitive functions with a preserved autonomy in daily life activities. MCI exhibits cognitive, behavioral, psychological symptoms. The executive functions (EFs) are key functions for everyday life and physical and mental health and allow for the behavior to adapt to external changes. Higher-level executive functions develop from basic EFs (inhibition, working memory, attentional control, and cognitive flexibility). They are planning, reasoning, problem solving, and fluid intelligence (Gf). This systematic review investigates the relationship between higher-level executive functions and healthy and pathological aging, assuming the role of executive functions deficits as a predictor of cognitive decline. The systematic review was conducted according to the PRISMA Statement. A total of 73 studies were identified. The results indicate that 65.8% of the studies confirm significant EFs alterations in MCI (56.8% planning, 50% reasoning, 100% problem solving, 71.4% fluid intelligence). These results seem to highlight a strong prevalence of higher-level executive functions deficits in MCI elderly than in healthy elderly.

## 1. Introduction

Mild Cognitive Impairment (MCI) is a syndrome characterized by a clinical profile intermediate between healthy aging and pathological aging. Individuals with MCI do not meet the diagnostic criteria of dementia, but they have worse cognitive functioning than physiological and normal aging [1]. The most common onset symptom is memory impairment, as in Alzheimer’s disease (AD), followed by other impairments [1]. However, cognitive deficits can be detected in cognitive functions other than memory. Petersen et al. [2] divided MCI into four groups based on the number and the type of impaired functions.

The most studied type is amnesic MCI, in which the subject has a memory disorder that can be at a single domain (aMCI) or multiple domains (aMCI–md) [3]. In the latter case, there are other impairments in addition to the memory deficits. On the other hand, if the subject does not have a memory deficit, we speak of non-amnestic MCI, which can be at a single (naMCI) or multiple (naMCI–md) domain based on the functions involved [2]. In 8–12% of cases, MCI evolves into Alzheimer’s disease. Hence, studying this syndrome is fundamental to predicting AD progression [2].

People with aMCI exhibit a reduced thickness of the entorhinal cortex, fusiform gyrus, and hippocampus compared to naMCI and healthy elderly, and reduced thickness of cingulate gyrus and amygdala compared to healthy elderly. A decreased thickness of precuneus is present in both MCI types [4]. These alterations are similar to the Alzheimer’s disease modifications, thus confirming how the MCI is a transitional phase between healthy and pathological aging from an anatomical point of view [5]. The patients with Mild Cognitive Impairment show behavioral and psychological symptoms, in addition to cognitive impairments involving memory and executive functions deficits [6].

The executive functions (EFs) are key functions for everyday life and physical and mental health, which allow adapting the behavior to external changes. The EFs are coordinated and integrated by different neural systems [7,8,9]. Executive functions deficits are the most common cognitive diseases, which can be found in several pathologies [10]. The Prefrontal Cortex (PFC) is the main area regulating EFs [7,9], and its damage can lead to the dysexecutive syndrome. This syndrome may be characterized by behavioral symptoms (e.g., social, sexual, and food behavior disorders, confabulation, and anosognosia) and/or cognitive symptoms (e.g., planning, shifting, theory of mind, and sustained attention disorders) [10]. Godefroy et al. [10] observe that planning is the most compromised cognitive ability in MCI and AD.

According to Diamond’s model, the EFs have four major components: inhibition, working memory, attentional control, and cognitive flexibility. Higher-level executive functions develop from these components. They are planning, reasoning, and problem-solving. Fluid intelligence (Gf) is considered a synonym of the latter two functions [7].

Planning is the ability to organize the behavior in certain situations, and think about the future to achieve a goal through a series of intermediate steps [11]. Shallice and Burgess [12] have described the planning process via four steps: (1) goal articulation; (2) plan formulation; (3) marker creation and triggering; (4) evaluation of initial goals achievement. The planning ability involves the right frontal area and the left frontal lobe. It requires sustained attention, inhibition of automatic responses [13], and the ability to hypothesize different scenarios and their consequences [14].

Reasoning is the ability to convert implicit information into explicit ones, clarify the process if necessary [15], and come to conclusions [16]. It can be divided into inductive and deductive reasoning, and the involved areas are left inferior and middle frontal gyrus, left middle and lateral temporal gyrus, left superior temporal, and cingulate gyrus [16]. Inductive reasoning involves the ability to make predictions regarding new situations based on pre-existent knowledge and is fundamental for categorization, scientific inference, probability, and decision-making [17]. Deductive reasoning is the ability to make logical inferences, and it is one of the major components of intelligence [18].

Problem solving is the capacity to achieve a goal through a sequence of cognitive operations (step by step) or insight [19]. Ordinary problem solving is a way to solve problems based on previous experience and uses well-known solutions. On the other hand, insight is an unconventional way to solve problems and is defined as an unconscious process [20]. Ordinary problem solving is regulated by the Frontoparietal Cognitive Control Network, which includes the middle frontal gyrus, inferior frontal gyrus, and inferior parietal lobule, and it involves a greater activation of Default Mode Network (DMN). Insight in problem solving is regulated by the same areas that regulate the ordinary problem solving with the addition of the anterior cingulate cortex and temporoparietal junction, and it involves a higher activation of visuospatial attention and visual perception areas [19].

Fluid Intelligence is the ability to solve problems via pre-existent acquired information [7,21]. The fluid intelligence is one of the two factors that derive from “general intelligence” together with crystallized intelligence (Gc). Crystallized intelligence is directly related to learning, while the fluid intelligence is the adaptation of the knowledge to a new context. Additionally, the two factors differ in their maximum peak: the crystallized intelligence peaks in early adulthood and is less affected by aging [21]; the fluid intelligence peaks in adolescence, is sensitive to physiological aging, and decreases more during this phase, but it can be stimulated by schooling, education, behavioral training, and stimulant drugs [21,22,23]. The most involved areas in Gf are the left anterior frontal lobe, inferior parietal lobule, and left frontoparietal regions, which are part of the Dorsal Attention Network (DAN). The DAN selects the internal stimuli based on goals or expectations and directs them to the appropriate cognitive or motor response [22,23].

This work proposes to analyze the executive functions in healthy elderly and Mild Cognitive Impairment since there is no other study of our knowledge that observes all the EFs together in these populations. Furthermore, the present study was carried out to evaluate how higher-level EFs could affect the everyday life of healthy and pathological elderly, and how they affect the independence in functional activity (IADL). IADL is one of the criteria of Mild Cognitive Impairment diagnosis [1], which is useful to distinguish it from dementia and could lead to an early diagnosis. This systematic review aims to investigate the relationship between higher-level executive functions and healthy and pathological aging, assuming the role of executive functions deficits as a predictor of cognitive decline. An additional goal is to establish what tests best discriminate healthy elderly from Mild Cognitive Impairment. Moreover, the present review points at evaluating how higher-level executive functions are compromised in both MCI and healthy elderly but with a worse outcome in pathological aging.

## 2. Materials and Methods

The review process was conducted according to the PRISMA Statement [24,25].

### 2.1. Research Strategies

A systematic search of the international literature was conducted in the following electronic databases by selecting articles published in peer-review journals: PsycINFO, Scopus, MEDLINE, and Web of Sciences. The last search was conducted on 13 July 2021. 

A list of keywords and MeSH terms was generated to identify studies (“Mild Cognitive Impairment” AND “executive function*”); (“Mild Cognitive Impairment” AND “planning”); (“Mild Cognitive Impairment” AND “reasoning”); (“Mild Cognitive Impairment” AND “problem solving”); (“Mild Cognitive Impairment” AND “fluid intelligence”). Restrictions were made, limiting the research to academic publications with English and Italian full text, without restrictions regarding gender and ethnicity. Additionally, the bibliographical references of retrieved papers, reviews, and meta-analyses were screened manually to assess whether they included relevant studies in the review. The number of selected articles is shown in Table 1.

### 2.2. Eligibility Criteria

A total of 11,269 articles were obtained from the search procedure. The first step allowed 5337 duplicates to be eliminated using the Mendeley software. Then, the list of potential articles produced by systematic research was revised. The reading of the title and abstract allowed the first exclusion of 5198 non-inherent studies. A further selection was made by reading the full text (See Figure 1).

The inclusion criteria were: adult population (age equal to or higher than 50 years), diagnosis of Mild Cognitive Impairment; healthy subjects; use of higher-level executive functions measurements.

The exclusion criteria were: participants with medical conditions that could potentially influence the investigated relationship (for example, metabolic disorders; cardiovascular disorders; chronic disorders; cancer); participants diagnosed with dementia (Alzheimer Disease; Parkinson’s Disease; Vascular Dementia; Frontotemporal Dementia; Dementia with Lewy Bodies; Huntington’s Disease), psychiatric disorders, neurological disorders, strokes; use of drugs that affect the nervous system and traumatic brain injury; methodological flaws; lack of essential data; assessment made by caregivers; MCI participants included in healthy elderly or AD groups; reviews, dissertations, editorials, comments, replies; trials; age < 50 years and animal models.

### 2.3. Data Collection

According to the PICOS approach [24], the following information was extracted from each study: authors and year of publication; characteristics of participants (including age, gender, Mini Mental State Examination—MMSE score); diagnostic criteria; experimental paradigm; results.

The extracted data are included in Table 2.

### 2.4. Quality Assessment

A quality assessment was carried out to analyze the eligibility of each article to reduce the risk bias. The analysis used five criteria to screen each study selected for systematic review: sampling bias, executive function measurements, diagnostic criteria, selective reporting bias, and methodological bias. Each criterion score ranges from 1 (low risk) to 3 (high risk). The overall quality shall be calculated by adding all the scores obtaining a global score ranging from 5 to 15. The study was considered at low risk of bias if the score was 5, while a score in the 6–10 interval was considered an indicator of a moderate risk of bias. The quality assessment was subdivided into planning, reasoning, fluid intelligence, and problem-solving measurements. The risk of bias is reported in Figure 2.

#### 2.4.1. Quality Assessment of Planning

Figure 3 shows the percentage of articles adopting planning tests fulfilling each quality criterion by the risk of bias assessment. On average, the quality of the studies was good since 36 out of 37 studies (97.3%) exhibited low scores on the risk of bias. The high percentage of studies with low or no risk of bias increases the validity of this systematic review. Despite one study (2.7%) showing moderate scores, no study reports a moderate risk of bias in more than two items. A large percentage of the studies adopted valid and reliable tools to measure planning and included an appropriate sample size. Moreover, most studies were adequately controlled for confounding variables. The higher risk bias was in the “EFs measurements” and the lower in “methodological bias”. In the overall bias, the score ranged from 5 to 7 for every article included.

#### 2.4.2. Quality Assessment of Reasoning

Figure 4 shows the percentage of articles adopting reasoning tests fulfilling each quality criterion of risk of bias assessment. On average, the quality of the studies was good since 28 out of 32 studies (87.5%) exhibited low scores on the risk of bias. The high percentage of studies with low or no risk of bias increases the validity of this systematic review. Despite four studies (12.5%) showing moderate scores, no study reports a moderate risk of bias in more than two items. A large percentage of the studies used valid and reliable tools to measure reasoning and included an appropriate sample size. Moreover, most studies were adequately controlled for confounding variables. The higher risk bias was in the “methodological bias” and the lower in “sampling bias” “EFs measurements” and “diagnostic criteria”. In the overall bias, the score ranged from 5 to 7 for every article included.

#### 2.4.3. Quality Assessment of Problem Solving

Figure 5 shows the percentage of articles adopting a problem-solving task fulfilling each quality criterion of risk of bias assessment. On average, the quality of the studies was good since six out of six studies (100%) exhibited low scores on the risk of bias. The high percentage of studies with low or no risk of bias increases the validity of this systematic review. No study reports a moderate risk of bias in more than one item. A large percentage of the studies used valid and reliable tools to measure problem solving and included an appropriate sample size. Moreover, most studies were adequately controlled for confounding variables. The higher risk bias was in the “EFs measurements” and the lower in “sampling bias”, “methodological bias”, and “diagnostic criteria”. In the overall bias, the score ranged from 5 to 6 for every article included.

#### 2.4.4. Quality Assessment of Fluid Intelligence

Figure 6 shows the percentage of articles adopting fluid intelligence measurements fulfilling each quality criterion of risk of bias assessment. On average, the quality of the studies was good since six out of seven studies (85.7%) exhibited low scores on the risk of bias. The high percentage of studies with low or no risk of bias increases the validity of this systematic review. Despite one study (14.3%) showing moderate scores, no study reports a moderate risk of bias in more than two items. A large percentage of the studies used valid and reliable tools to measure fluid intelligence and included an appropriate sample size. Moreover, most studies were adequately controlled for confounding variables. The higher risk bias was in the “methodological bias” and the lower in “EFs measurements” and “diagnostic criteria”. In the overall bias, the score ranged from 5 to 7 for every article included.

## 3. Results

### 3.1. Studies Selection

The flow chart shows the number of studies identified from the databases and the number of studies examined, assessed for eligibility, and included in the review with the reasons for possible exclusions (see Figure 1). A total of 73 studies were identified.

Of the 73 selected studies, 30 analyzed planning, 31 reasoning, six problem solving, and seven fluid intelligence. Nine studies used different executive function measures.

Results will be presented in two subsections, according to the higher-level executive functions and the MCI subtype.

### 3.2. Planning (N = 37)

Thirty-seven studies have measured planning in healthy elderly and MCI participants with an overall sample of 3491 participants (1919 HC and 1572 MCI) with a mean age that ranges from 60.7 years [82] to 79 years [90].

Thirteen studies used “CLOX-1” or “Clock Drawing Test (CDT)” [56,83,84,89,90,94,100,105,107,113,115,116,117]; two studies used the “Wisconsin Card Sorting Test–Computer Version (WCST–CV)” [81,82]; 10 studies used the “Tower of London (TOL)” [28,33,46,54,73,74,91,92,95,96], six studies used “Zoo Map Test” [29,53,64,98,99,111], two studies used “Tower of Hanoi (TOH)” [36,108]; one study used “Raven’s Coloured Progressive Matrices (RCPM)” [26]; one study used “Tower Test (D-KEFS)” [71]; one study used “Groton Maze Learning Test” [88]; one study used “Trail Making Test (B-A)” [114]; one study used “Porteus Maze Test” [114]; one study used “Verbal Fluency Test (fruits and animals version)” [114]; one study used “Action Program Test” [53]; and one study used “Key search Test” [53].

Fifteen studies did not report any significant difference between groups [26,28,29,56,83,84,89,90,95,105,107,108,110,115,116,117]. One study [53] performed three tests to assess planning ability and observed a worse performance in MCI in only two of them (Action Program Test and Zoo Map Test). The remaining 21 studies reported poor performance in MCI than healthy subjects [33,36,46,54,64,71,73,74,81,82,88,89,90,94,97,98,99,100,113,114].

Nine studies [28,36,46,54,71,73,74,92,97] of the thirteen that analyzed the planning abilities with tower tests (“Tower of London”, “Tower of Hanoi”, and “Tower Test (D-KEFS)”) highlighted a poorer performance in MCI than healthy groups. Metzler-Baddeley et al. [74] and Berlot et al. [33] observed more rule violations during the performance of the task in MCI, while Bharath et al. [36] reported a longer time to complete the test. De Paula et al. [46] used two versions of “Tower of London” (designed by Portella et al. [47] and Krikorian et al. [48]) and observed a lower planning ability in MCI subjects. Rainville et al. [92] pointed out a higher rule breaking and abandonment rate in MCI. Sánchez–Benavides et al. [96] saw in Mild Cognitive Impairment subjects higher total moves, total initiation time, total exclusion time, total solving time, and lower total correct rates than healthy subjects. Also, Garcia-Alvarez et al. [54], Lindbergh et al. [71], and Lussier et al. [73] found poor planning in Mild Cognitive Impairment subjects.

Four studies [89,94,100,113] that used “CLOX-1” found a lower planning capacity in Mild Cognitive Impairment samples. 

Three studies used the “Zoo Map Test” [64,98,99]. Sanders et al. [98] highlighted higher total errors in MCI than healthy controls. Junquera et al. [64] analyzed the differences between healthy subjects, aMCI, naMCI, and aMCI multiple domains: aMCI multiple domains showed lower planning than healthy elderly and aMCI single domain, while naMCI subjects showed a poor planning ability than healthy elderly. Sanders et al. [98] observed decreased planning ability in MCI compared to healthy elderly.

Espinosa et al., [53] analyzed the differences between healthy subjects and MCI using the “Zoo Map Test” and “Action Program Test”, in both tests, MCI had lower planning than healthy controls. Nordlund et al. [81,82] found a poor planning ability, assessed with the “Wisconsin Card Sorting Test–Computer Version (WCST–CV)”, in MCI subjects than healthy elderly. Papp et al. [88] used the “Groton Maze Learning Test” to evaluate planning and underlined that participants with MCI exhibit higher exploratory errors, more rule-breaks errors, and reduced differences between trial 1 and trial 2 than healthy subjects. Zhang et al. [114] evaluated the differences between healthy subjects and MCI with “Trail Making Test (B-A)”, “Porteus Maze Test”, and “Verbal Fluency (fruits and animals)”, and in each test, Mild Cognitive Impairment showed lower planning than healthy elderly.

### 3.3. Reasoning (N = 32)

Thirty-two studies measured reasoning in healthy elderly and Mild Cognitive Impairment subjects, with an overall sample of 3371 participants (1676 HC and 1695 MCI) and a mean age ranging from 60.7 years [82] to 82 years [55].

Seventeen studies used “Similarities” [42,50,60,61,62,64,65,66,68,70,77,80,81,82,85,106,109]; six studies used “Matrix Reasoning” [41,42,44,58,62,85]; four studies used “Raven’s Coloured Progressive Matrices (RCPM)” [37,78,101,102]; four studies used “Proverbs” [60,61,65,75]; two studies used “Cognitive Competency Test” [84,105]; two studies used “Abstraction–MoCA” [55,115]; two studies used “DRS–2 Conceptualization” [84,105]; one study used “ACED Money Management” [72]; one study used “MacCAT-T” [72]; one study used “Block Design” [49]; one study used “Mattis Identities and Oddities” [105]; one study used “Raven’s Progressive Matrices (RPM)” [32]; one study used “Abstraction (Wechsler et al., [118], Kramer et al., [119])” [87].

Fifteen studies did not report any significant difference between samples [49,58,60,61,65,77,80,81,82,84,85,87,101,102,115].

Three studies [42,72,106] performed two tasks each and observed lower reasoning ability in MCI participants in only one of them.

The remaining 14 studies reported differences between MCI and healthy subjects [32,37,41,44,55,62,64,66,68,70,75,78,105,109].

Seven studies used “Similarities” [62,64,66,68,70,106,109] to assess reasoning in healthy elderly and Mild Cognitive Impairment samples and reported lower performance in reasoning in MCI subjects. Junquera et al. [64] analyzed the differences between healthy subjects, aMCI, naMCI, and aMCI multiple domains: both aMCI multiple domains and naMCI showed lower reasoning than healthy elderly. Four studies used “Matrix Reasoning” [41,42,44,62] to evaluate MCI and healthy subjects, and in each study, a decreased reasoning in Mild Cognitive Impairment subjects was highlighted. In particular, Chang [41] observed a higher performance in healthy subjects than MCI with normal awareness for memory (MCI-na), which in turn were better than MCI with poor awareness for memory (MCI-pa). Two studies used “Raven’s Coloured Progressive Matrices–RCPM” [37,78] and observed a reduced reasoning ability in Mild Cognitive Impairment participants, as well as Benavides-Varela et al., [32] that used “Raven’s Progressive Matrices–RPM”. Sherod et al. [105] analyzed the differences between healthy subjects and MCI with the “DRS–2 Conceptualization” and “Cognitive Competency Test”, and in both tests MCI had lower abstraction than healthy controls. Lui et al., [72] used “ACED Money Management” and reported a reduced reasoning ability in Mild Cognitive Impairment. Moreira et al. [75] used “Proverbs” to evaluate the reasoning in healthy elderly and MCI participants and observed higher abstraction ability in healthy subjects than MCI. García et al. [55] used “Abstraction (MoCA)” and pointed out a reduced ability in abstraction in MCI subjects.

### 3.4. Problem Solving (N = 6)

Six studies have assessed problem solving in MCI and a control group, with an overall sample composed of 344 participants (236 MCI and 108 HC) and a mean age ranging from 62.6 years [63] to 82 years [39]. Each study [35,51,63,88,104] used a different task to evaluate problem solving, and they all showed differences between the samples.

Beversdorf et al., [35] used the “Matchstick Problem” to evaluate visuospatial problem solving and highlighted lower capacity in the MCI sample to solve problems. Burton et al., [39] used “Block Design” to evaluate problem solving ability in healthy elderly, aMCI single and multiple domains, naMCI single and multiple domain participants. They observed better performance in healthy subjects than amnesic Mild Cognitive Impairment multiple domains, non-amnesic Mild Cognitive Impairment single, and multiple domains, but not to aMCI single domain subjects. Dwolatzky et al., [51] used “Pictorial Puzzles (2x2)” and reported a reduced accuracy in Mild Cognitive Impairment. Jin et al., [63] evaluated problem solving with “Sudoku (Nikoli Publishing)” and highlighted a decreased accuracy in complex tasks in aMCI subjects compared to healthy subjects. Papp et al., [88] used the “Groton Maze Learning Test” to evaluate problem solving and underlined that participants with MCI have higher exploratory errors, more rule-breaks errors, and a lower difference between trial 1 and trial 2 than healthy subjects. Sheldon et al., [104] used the “Means-Ends Problem Solving Test” and observed a reduced problem-solving ability in MCI subjects.

### 3.5. Fluid Intelligence (N = 7)

Seven studies have measured fluid intelligence in healthy elderly and Mild Cognitive Impairment subjects, with an overall sample of 682 participants (341 HC and 341 MCI) and a mean age ranging from 67.37 years [103] to 75.3 years [69]. Five studies used “Block Design” [45,58,66,69,112], three studies used “Matrix Reasoning” [52,103,112], and one study used “Raven Coloured Matrices” to evaluate fluid intelligence.

Two studies [58,66] did not report any significant difference between samples, one study performed multiple tests and showed conflicting results [110], while the others [45,52,69,103] reported lower performance in fluid intelligence in MCI subjects.

### 3.6. Amnesic Mild Cognitive Impairment (N = 20)

Twenty studies analyzed the differences between aMCI and healthy elderly in higher-level executive functions, with an overall sample of 1783 participants (969 HC and 814 aMCI) and a mean age ranging from 62.6 years [64] to 82 years [39]. Eleven of these (55%) reported a significant difference between healthy and pathological elderly. Eight studies [26,28,64,69,88,115,116,117] compared planning ability in aMCI and healthy elderly, and only three of these (37.5%) [64,69,88] reported a worse performance in Mild Cognitive Impairment. Ten studies [44,58,64,65,68,70,85,101,102,115] analyzed reasoning in amnesic MCI and a control group, four of these (40%) [44,64,68,70] highlighted a poorer reasoning ability in MCI. Four studies [39,63,88,104] investigated problem solving in amnesic Mild Cognitive Impairment and healthy control, and all (100%) of them highlighted a significant difference between groups. Two studies [58,112] evaluated fluid intelligence, and only one [112] observed a lower Gf in aMCI. Additionally, only four [28,39,44,64] of these studies have distinguished between amnesic Mild Cognitive Impairment single and multiple domains. In two of them [44,64], aMCI+ have reported a poorer performance regarding aMCI single domain with respect to healthy subjects.

### 3.7. Non-Amnesic Mild Cognitive Impairment (N = 4)

Four studies analyzed the differences between naMCI and healthy elderly in higher-level executive functions, with an overall sample of 495 participants (360 HC and 135 naMCI) and a mean age ranging from 63.4 years [102] to 79.57 years [39]. Three of these (75%) reported a significant difference between groups. One study [64] evaluated planning in naMCI and healthy elderly, highlighting a poorer performance in MCI sample. The most analyzed higher-level executive function in non-amnesic MCI is reasoning, which is evaluated in three studies [64,70,102], but only one (33.3%) [64] of them highlighted a lower reasoning ability in naMCI. Finally, one study [39] analyzed problem solving and observed a worse performance in naMCI compared to healthy elderly. Only one study [39] set apart non-amnesic Mild Cognitive Impairment multiple domain from naMCI single domain, showing no difference between the two groups. No one evaluated fluid intelligence in non-amnesic Mild Cognitive Impairment, and no one analyzed them without the amnesic Mild Cognitive Impairment sample.

## 4. Discussion

The purpose of this systematic review was to investigate the relationship between higher-level executive functions and healthy and pathological aging, assuming the role of executive functions deficits as a predictor of the general cognitive decline. Results showed that not all the studies found a prevalence of higher-level executive functions deficits in individuals with Mild Cognitive Impairment diagnosis compared to healthy elderly; however, 64.4% of the studies confirm a significant presence of alterations in MCI (56.8% planning, 50% reasoning, 100% problem solving, 71.4% fluid intelligence).

Despite the scarce number of observations that do not allow reliable conclusions, the evaluation of problem solving showed significant results. These data must be interpreted with caution because the studies [35,39,51,63,88,104] used different tasks to evaluate this ability. One interesting finding was observed by Burton et al. [39] that compared healthy subjects, amnesic Mild Cognitive Impairment single and multiple domains, and non-amnesic Mild Cognitive Impairment single and multiple domains to analyze problem solving. According to the literature, the authors reported lower problem-solving capacity in participants with aMCI multiple domains and naMCI single and multiple domains compared to healthy elderly, while the aMCI single domain subjects did not report any significant difference with the others. These results could be attributed to the Mild Cognitive Impairment [1], in which the only impaired cognitive domain is memory. On the other hand, Jin et al. [63] found a significant difference between aMCI and healthy control group; the author reported a positive linear correlation between blood oxygen levels in the posterior cingulate cortex (PCC) and precuneus in aMCI subjects during simple (r = 0.95) and complex (r = 0.90) problem solving tasks. In addition, healthy elderly showed a deactivation of these areas while the aMCI showed an activation. These regions are included in the Default Mode Network (DMN) and, taking into account the close relationship with the hippocampus, these activations in aMCI may be explained as a compensatory memory mechanism.

The results of fluid intelligence must be interpreted with caution due to the small number of studies that measured this variable [45,52,58,66,69,103,112]. A possible source of error about fluid intelligence ability is linked to the type of assessment carried out: this review includes studies that evaluated the intelligence quotient employing tests commonly used to assess fluid intelligence.

Despite this, Li et al., [69], through the means regression and the cluster analysis, observed that the “Block Design” test could predict conversion from a healthy state to amnesic Mild Cognitive Impairment. Another important finding is observed by Wu et al., [111] that studied, in amnesic Mild Cognitive Impairment and healthy elderly, the Resting State-Executive Control Network (RS-ECN), a network that is adjacent to DMN and the other major attention networks and with which it shares some anatomical areas. The aMCI showed a decreased functional connectivity of anterior cingulate cortex (ACC), inferior parietal lobule (IPC), lateral parietal and anterior insula, precuneus, middle frontal gyrus, left and right dorsal lateral prefrontal cortex (DLPFC); these regions are strictly involved in the Ventral Attention Network and more generally in executive functions. Moreover, the author [112] also observed increased functional connectivity of different areas of the Default Mode Network, the Ventral Attention Network (VAN), and the Dorsal Attention Network: the right anterior prefrontal cortex (aPFC), left and right ventral lateral prefrontal cortex (VLPFC), superior parietal cortex, posterior parietal lobule, occipital and temporal. Even if these regions are not involved in fluid intelligence, they are still implicated in planning, reasoning, problem solving, abstract thinking, and other executive functions, with particular reference to the DLPFC. The overall results of fluid intelligence, although not uniform, pursued a trend towards higher prevalence of this ability deficit in Mild Cognitive Impairment.

The results about planning are projected to highlight a negative trend in MCI that reported lower ability than healthy elderly. However, these data must be interpreted with caution because not all tests provided statistically significant results, and some studies used inappropriate tests. In particular, some studies used the “Clock Drawing Test” and “CLOX-1”, which are not specific for planning evaluation but are instead typically used in neuropsychological batteries to investigate other cognitive functions. The “Clock Drawing Test” is commonly used to assess praxis and visuospatial skills, while the “CLOX-1” is the version that evaluates the executive functions (e.g., goal selection, planning, selective attention, and motor sequencing) [120]. Despite this distinction, four studies [57,116,117,118] used the “Clock Drawing Test” to assess executive functions, and neither of these reported any significant difference between MCI and healthy elderly. In addition, the studies that used “CLOX–1” [83,84,89,104,106] did not report significant differences between MCI and healthy subjects; only a few of studies [89,94,100,113] highlighted lower planning ability in pathological aging. These results could be explained by the low sensitivity of this test in discriminating between MCI and healthy elderly. However, the studies that used the “Clock Drawing Test” and the “CLOX-1” were included too, since both original validations [120,121] considered the test adequate to assess planning ability.

On the other hand, the “Zoo Map Test” and the tower tests (“Tower of London”, “Tower of Hanoi”, and “Tower Test (D-KEFS)”) seem well to discriminate the differences between healthy and MCI participants. Junquera et al., [64] analyzed the differences between healthy subjects and single and multiple domain aMCI and naMCI participants. Subjects with aMCI multiple domains and single and multiple domains naMCI subjects exhibited lower planning ability than healthy elderly. Two studies [33,74] observed more rule violations during tasks in MCI; in addition, Rainville et al., [92] pointed out a higher rule breaking and abandonment rate in MCI than healthy participants. Metzler-Baddeley et al., [74] have also observed a correlation between the number of rule violations in the TOL and the variation of mean diffusivity in the bilateral anterior cingulum and the fornix.

Although not all results showed a statistically significant difference, many reasoning deficits can be observed in MCI. Most studies used the “Similarities” test to evaluate reasoning, which identifies the relationship between a couple of words. Seven of these studies [62,64,66,68,70,106,109] reported significant differences between healthy and Mild Cognitive Impairment elderly. Chang [41] compared healthy and MCI participants with and without awareness for memory problems and observed a higher performance in healthy subjects than in MCI with normal awareness for memory (MCI-na), which in turn were better than MCI with poor awareness for memory (MCI-pa). In addition, MCI-pa showed reduced white matter integrity of left dorsal frontal–striatal tract, right dorsal frontal–striatal tract, left anterior thalamocortical radiations–ventral part, corpus callosum–inferior parietal lobule, and corpus callosum–ventral prefrontal regions. Nishi et al., [78] found a correlation between reasoning task execution and reduced glucose reuptake in the right middle frontal gyrus and higher activation in the same area.

Not all the studies that analyzed higher-level executive functions highlighted significant differences. Generally, it may be concluded that elderly with Mild Cognitive Impairment exhibit poorer performance than healthy elderly. Due to small observations, problem solving and fluid intelligence results do not allow reliable conclusions. Despite this, the results appear promising, showing higher executive function deficits in MCI. Though numerous and highlighting a worse performance in MCI, planning and reasoning results do not always show significant differences between groups. This could be related to low sensitivity measures to discriminate MCI from normal aging.

### Limits

Despite the encouraging results, this review holds some limitations. The major limitation is the lack of quantitative analysis (meta-analysis), which is difficult to carry out because of the large number of different tests and diagnostic criteria adopted by the studies. The absence of a standardized protocol to evaluate the higher-level executive functions represents another limitation, leading to the administration of rarely used tests and, consequently, to hardly generalizable results. An additional limiting factor of this review is task impurity and, therefore, the difficulty of separately evaluating each higher-level EF. For example, the “Matrix Reasoning” is used to evaluate: reasoning [42,44,85], visuospatial reasoning [58], non-verbal abstract reasoning [62], intelligence quotient [103,111], and fluid intelligence [52]. A further limit can be related to publication bias. Lastly, this review is based on Diamond’s model [7], and therefore it focuses on some executive functions excluding all others, such as decision-making.

## 5. Conclusions

The results of this systematic review seem to highlight a higher prevalence of higher-level executive functions disease in elderly with Mild Cognitive Impairment than in healthy elderly, confirming results already observed with other executive functions, such as cognitive and motor inhibition, conflict control, and cognitive flexibility [122], although some of these EFs are also compromised in healthy elderly [123]. MCI shows modifications over every aspect investigated in this research, highlighting significant differences that could worsen the quality of life. As far as we know, this study is the first to evaluate these aspects in healthy and MCI elderly. Certainly, a future goal will be to establish and create a standardized protocol to discriminate MCI from healthy elderly. Such a protocol should accurately measure reasoning, planning, problem solving, and fluid intelligence since these functions are treated as a single construct included in executive functions. An important goal for the next studies will be to figure out if higher-level executive functions diseases are early symptoms of Mild Cognitive Impairment or, on the other hand, MCI leads to poorer higher-level executive functions abilities as a consequence of the more significant alterations of the nervous system occurring in pathologically older age than in healthy elderly.

## Figures and Tables

**Figure 1 jcm-11-01204-f001:**
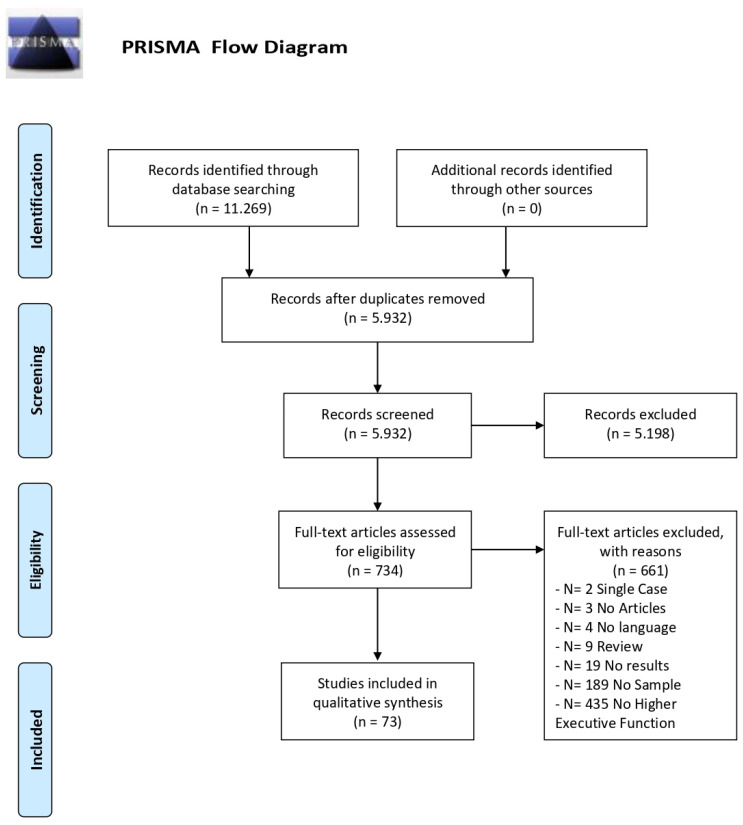
Flow diagram, PRISMA Statement [24,25].

**Figure 2 jcm-11-01204-f002:**
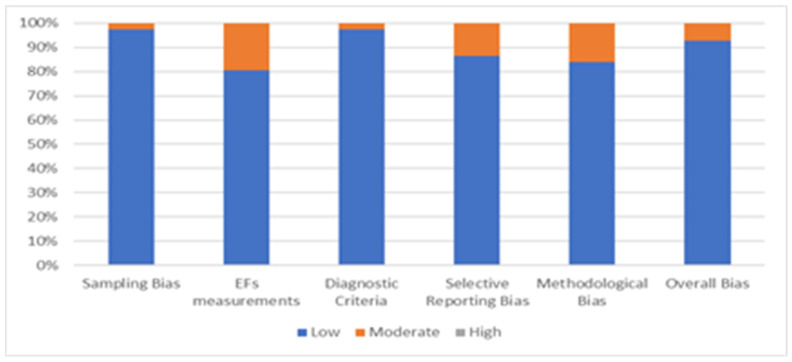
Risk Bias of selected articles, considering all higher executive functions.

**Figure 3 jcm-11-01204-f003:**
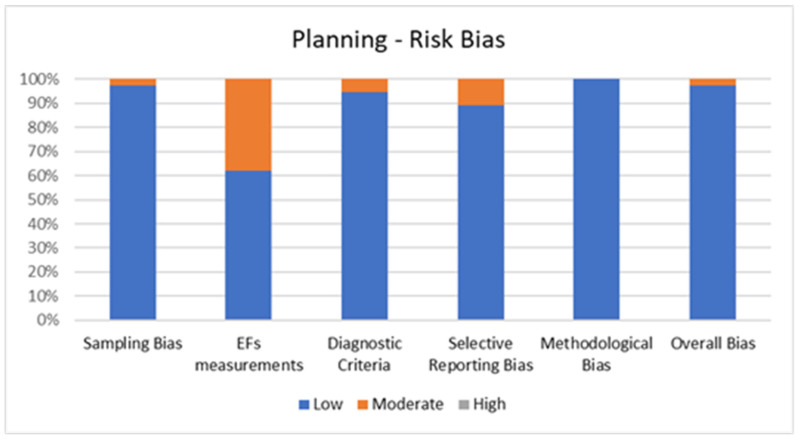
Risk of Bias for Planning.

**Figure 4 jcm-11-01204-f004:**
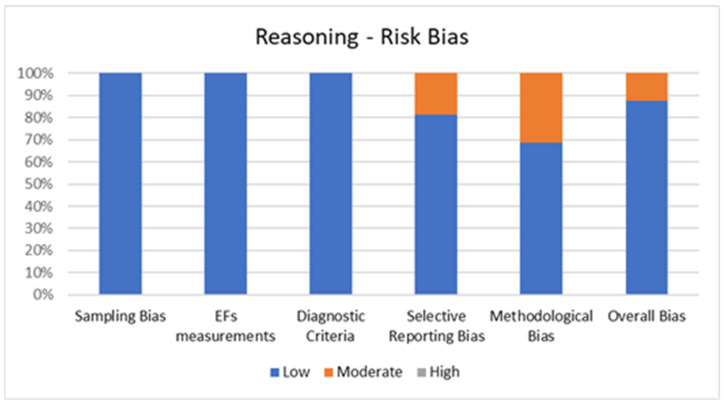
Risk of Bias for Reasoning.

**Figure 5 jcm-11-01204-f005:**
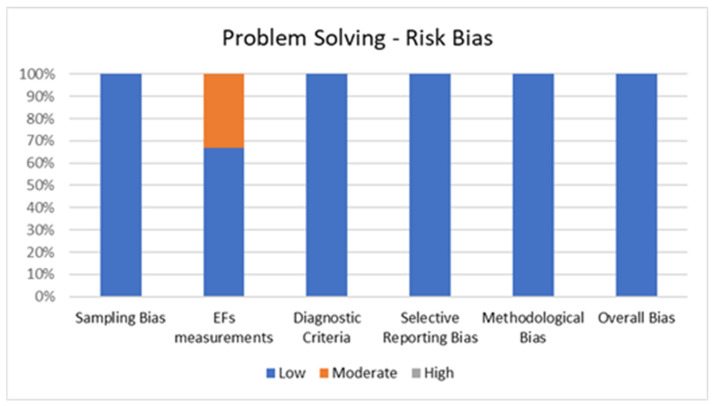
Risk of Bias for Problem Solving.

**Figure 6 jcm-11-01204-f006:**
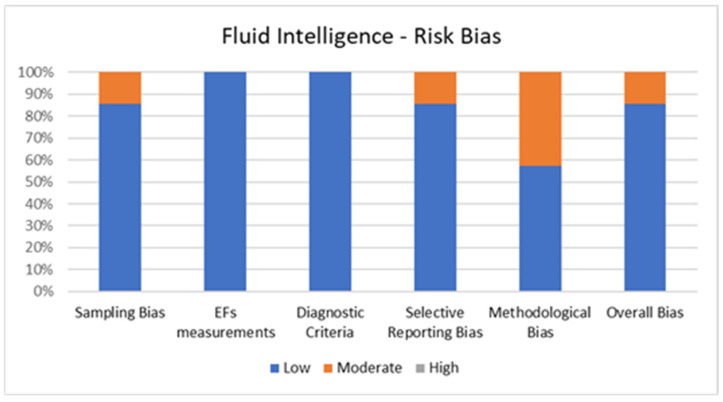
Risk of Bias for Fluid Intelligence.

**Table 1 jcm-11-01204-t001:** Number of selected articles in databases.

Database	N°
PsychINFO	1581
MEDLINE	4067
Scopus	2881
Web of Sciences	2740

**Table 2 jcm-11-01204-t002:** Results of selected studies.

Authors	Group	N°	Age (SD)	(%F)	MMSE (SD)	Diagnostic Criteria	Test	Results
Ambra et al. [26]	aMCI	15	69.4 (7.59)	33.33	-	[27]	RCPM	No difference in planning has been found
	HC	31	69.2 (7.2)	45.16	-			
Ávila et al. [28]	HC	26	70.58 (7.17)	-	26.85 (3.04)	[6]	TOL	No difference in planning has been found
	aMCI	38	73.03 (7)	-	26.58 (2.03)			
	aMCI+	29	77 (7.43)	-	23.52 (3.17)			
Beaver et al. [29]	HC	65	72.34 (8.78)	63.1	-	[30,31]	Zoo Map Test	No difference in planning has been found
	MCI	19	70.53 (9.35)	52.6	-			
	MCI+	33	71.37 (8.39)	48.5	-			
Benavides-Varela et al. [32]	MCI	43	75.44	42	26.39 (2.84)	[30]	RPM	MCI: lower abstract reasoning than HC
	HC	37	68.89	46	28.73 (1.17)			
Berlot et al. [33]	HC	20	74 (6.5)	50	-	[34]	Tower Test (D-KEFS)	MCI: higher rule violations than HC
	MCI	25	76.8 (7.3)	44	-			
Beversdorf et al. [35]	MCI	26	67.5 (8.9)	53.85	26.1 (1.7)	MMSE > 24 CDR = 0.5	Matchstick Problem	MCI: lower visuo-spatial problem solving than HC
	HC	20	68.0 (8.3)	70	28.8 (1.4)			
Bharat et al. [36]	MCI	56	68.76 (7.59)	30.4	27.74 (2.43)	[1]	TOH	MCI: higher time than HC
	HC	59	67.13 (5.62)	32.3	30.83 (0.64)			
Borella et al. [37]	MCI	15	72.73 (5.28)	60	27.4 (1.45)	[1,38]	RCPM	MCI: lower logical reasoning than HC
	HC	18	69.72 (3.20)	61.11	29.5 (0.62)			
Burton et al. [39]	HC	158	73.57 (4.72)	-	28.92 (1.17)	[6,40]	Block Design	HC: performed better than naMCI, naMCI+ and aMCI+
	aMCI	6	79.5 (5.65)	-	26.83 (2.48)			
	naMCI	39	77.54 (5.61)	-	28.67 (2.48)			
	aMCI+	19	82 (5.04)	-	28.16 (1.26)			
	naMCI+	28	79.57 (4.86)	-	28.68 (1.09)			
Chang et al. [41]	HC	36	69.33 (4.09)	58.33	-	[31]	Matrix Reasoning	MCI-pa < MCI-na < HC
	MCI-na	24	71.54 (8.85)	58.33	-			
	MCI-pa	22	72.82 (7.83)	50	-			
Chao et al. [42]	HC	65	68.69 (6.8)	61.6	-	[6,43]	Matrix Reasoning Similarities	Matrix Reasoning MCI: lower reasoning than HC Similarities No difference in reasoning has been found
	MCI	54	73.46 (9.3)	54.6	-			
Chow et al. [44]	HC	52	75.19 (6.4)	48.07	-	[40]	Matrix Reasoning	aMCI+: lower reasoning than HC
	aMCI	34	76.41 (6.42)	58.82	-			
	aMCI+	20	79.15 (5.57)	30	-			
De Oliveira et al. [45]	HC	61	70.66 (6.55)	57.37	28.38 (1.48)	[6]	Block Design RCPM	Block Design MCI: lower fluid intelligence than HC RCPM MCI: lower fluid intelligence than HC
	MCI	38	72.32 (7.94)	63.15	25.79 (2.74)			
De Paula et al. [46]	MCI	60	73.7 (8.9)	53.33	24.23 (3.43)	[30]	TOL (Portella [47] and Krikorian version [48])	Portella et al., [47] MCI: lower planning than HC Krikorian et al., [48] MCI: lower planning than HC
	HC	60	74.1 (5.6)	55	27.08 (2.96)			
Djordjevic et al. [49]	HC	33	73.7	48.5	28.7	[38,50]	Similarities Block Design	Similarities No difference in verbal abstract reasoning has been found Block Design No difference in nonverbal reasoning has been found
	MCI	51	75.4	51	27.26			
Dwolatzky et al. [51]	HC	39	73.41 (8.0)	66.67	29.03 (1.11)	[38]	Pictorial Puzzles 2x2	MCI: lower accuracy in problem solving task than HC
	MCI	30	77.15 (6.43)	43.33	27.63 (1.54)			
Econoumou et al. [52]	MCI	31	73.58 (6.17)	-	28.10 (1.47)	[38]	Matrix Reasoning	MCI: lower fluid intelligence than HC
	HC	27	70.56 (8.87)	-	-			
Espinosa et al. [53]	HC	50	72.26 (7.85)	74	28.38 (1.68)	[38]	Action Program Test Key Search Test Zoo Map Test	Action Program Test MCI: lower planning than HC Key Search Test No difference in planning has been found Zoo Map Test MCI: lower planning than HC
	MCI	50	74.30 (6.93)	44	26.06 (2.68)			
Garcia–Alvarez et al. [54]	HC	124	73.17 (8.6)	60.48	28.49 (1.4)	[1]	TOL	MCI: lower planning than HC
	MCI	48	76.68 (10.3)	43.75	25.96 (2.03)			
García et al. [55]	MCI	5	82 (6.38)	40	24 (1.41)	Memory impairment; normal daily living; no dementia	Abstraction	MCI: lower abstraction than HC
	HC	5	74.25 (6.86)	40	28.25 (2.06)			
Griffith et al. [56]	HC	21	66.7 (7.2)	66.67	29.3 (1.0)	[38,57]	CLOX–1	No difference in planning has been found
	MCI	21	68.1 (8.8)	52.38	28.4 (1.2)			
Guild et al. [58]	HC	48	70.65 (4.47)	54.17	28.88 (1.36)	[59]	Block Design Matrix Reasoning	Block Design No difference in IQ has been found Matrix Reasoning No difference in visuo-spatial reasoning has been found
	aMCI	14	73.07 (6.44)	85.71	28.14 (1.46)			
Hellmuth et al. [60]	HC	41	68.2 (7.2)	68.29	29.6 (0.6)	CDR ≥ 0.5	3 Similarities and 3 Proverbs	No difference in abstraction has been found
	MCI	10	68.9 (8.8)	50	28.6 (1.8)			
Heuer et al. [61]	HC	118	69.4 (0.57)	58.47	29.54 (0.64)	CDR ≥ 0.5	3 Similarities and 3 Proverbs	MCI: lower abstraction than HC
	MCI	36	72.9 (1.12)	50	28.77 (0.24)			
Jefferson et al. [62]	HC	40	72.3 (5.5)	60	29.3 (0.9)	[1,6]	Similarities Matrix Reasoning	Similarities MCI: lower verbal abstract reasoning than HC Matrix Reasoning MCI: lower nonverbal abstract reasoning than HC
	MCI	40	74.3 (7.5)	48	27.8 (1.8)			
Jin et al. [63]	HC	13	62.6 (7.0)	30.77	29.1 (0.6)	MMSE > 24	Sudoku	MCI: lower accuracy in problem solving complex task
	aMCI	13	63.6 (7.8)	30.77	25.9 (1.8)			
Junquera et al. [64]	HC	51	71.2 (4.5)	-	28.94 (1.36)	[1,6]	Zoo Maps Test Similarities	Zoo Maps Test aMCI+: lower planning than HC and aMCI naMCI: lower planning than HC Similarities aMCI+: lower reasoning than HC naMCI: lower reasoning than HC
	aMCI	26	74.73 (4.53)	-	28.54 (1.27)			
	aMCI+	50	75.61 (6.46)	-	26.20 (2.99)			
	naMCI+	18	72.24 (6.14)	-	27.77 (2.45)			
Kramer et al. [65]	HC	35	73.0 (5.3)	-	29.5 (0.8)	[38]	2 similarities and 2 proverbs	No difference in abstract reasoning has been found
	aMCI	86	75.0 (6.1)	-	28.5 (1.5)			
Levinoff et al. [66]	HC	40	74.1 (7.1)	-	28.7 (1.2)	[67]	Similarities Block Design	Similarities MCI: lower in abstract verbal reasoning than HC Block Design No difference in fluid intelligence
	MCI	73	74.0 (7.3)	-	27.7 (1.9)			
Li et al. [68]	HC	28	71.25 (6.43)	60.71	27.61 (1.95)	[1,30]	Similarities	aMCI: lower abstract reasoning than HC
	aMCI	29	73.76 (6.42)	62.07	26.07 (2.33)			
Li et al. [69]	HC	111	73.56 (8.62)	65.8	26.0 (4.44)	CDR = 0	Block Design	aMCI: lower planning than HC
	aMCI	111	75.30 (7.12)	66.7	25.28 (3.47)			
Li et al. [70]	HC	123	66.26 (9.96)	69.1	28.5 (1.42)	[1]	Similarities	aMCI: lower abstract reasoning than HC
	aMCI	106	74.24 (8.05)	48.6	26.03 (2.6)			
	naMCI	37	71.46 (9.63)	67.6	27.35 (2.20)			
Lindbergh et al. [71]	HC	35	74.7 (5.97)	66.7	-	[30,34]	Tower test (D-KEFS)	MCI: lower planning than HC
	MCI	25	78.6 (5.22)	92	-			
Lui et al. [72]	HC	93	74.2 (6.5)	85.25	26.6 (2.5)	[30]	ACED money management MacCAT-T	ACED MCI: lower reasoning than HC MacCAT-T No difference in reasoning has been found
	MCI	92	77.8 (6.8)	71.74	25.3 (2.6)			
Lussier et al. [73]	HC	26	72.0 (6.4)	69	-	[1,6]	TOL	MCI: lower planning than HC
	MCI	22	75.8 (6.5)	36	-			
Metzler-Baddeley et al. [74]	HC	20	74.0 (6.5)	50	-	[34]	TOL	MCI: higher rule violation than HC
	MCI	46	76.8 (7.3)	44	-			
Moreira et al. [75]	HC	26	68.42 (8.39)	61.54	29.62 (0.7)	[76]	Proverbs	MCI: lower abstraction than HC
	MCI	32	68.03 (7.29)	46.87	27.69 (1.31)			
Muñoz-Neira et al. [77]	HC	30	71.93 (7.06)	50	28.77 (1.14)	[6]	Similarities	No difference in abstraction has been found
	MCI	14	71.71 (7.16)	42.9	26.29 (2.13)			
Nishi et al. [78]	MCI	30	69.8 (7.3)	73.33	26.5 (2.1)	MMSE ≥ 24 CDR = 0.5 NINCDS-ADRDA [79]	RCPM	MCI: lower reasoning than HC
	HC	15	70.9 (4.2)	60	29.1 (1.6)			
Nordlund et al. [80]	HC	112	67.0 (5.5)	-	29.3 (1.1)	MMSE < 25	Similarities	No difference in verbal abstraction has been found
	MCI	35	64.0 (8.2)	-	28.5 (1.5)			
Nordlund et al. [81]	HC	60	66.5 (6.2)	46.67	29.3 (1.1)	MMSE < 25	WCST-CV Similarities	WCST-CV MCI: lower planning than HC Similarities No difference in abstraction has been found
	MCI	60	66.4 (6.8)	46.67	28.4 (1.3)			
Nordlund et al. [82]	HC	50	65.1 (6.1)	54	29.3 (1.0)	MMSE < 25	WCST-CV Similarities	WCST-CV MCI: lower planning than HC Similarities No difference in abstraction has been found
	MCI	73	60.7 (6.8)	52.05	28.6 (1.3)			
Okonkwo et al. [83]	HC	43	66.76 (7.40)	62.79	29.38 (0.89)	[1]	CLOX–1	No difference in planning has been found
	MCI	43	69.54 (8.22)	44.19	28.54 (1.46)			
Okonkwo et al. [84]	HC	56	64.63 (8.5)	67.9	29.55 (0.76)	[30]	CLOX–1 DRS–2 Conceptualization Cognitive Competency Test	CLOX–1 No difference in planning has been found DRS–2 Conceptualization No difference in abstraction has been found Cognitive Competency Test No difference in verbal reasoning has been found
	MCI	60	68.05 (6.77)	56.7	28.37 (1.5)			
Pa et al. [85]	HC	36	64.8 (8.2)	63.89	29.8 (0.6)	[86]	Matrix Reasoning Similarities	Matrix Reasoning No difference in reasoning has been found Similarities No difference in reasoning has been found
	aMCI	26	68.0 (6.6)	50	28.7 (1.2)			
Pa et al., [87]	MCI	57	69.8 (9.3)	47.37	28.4 (1.5)	[6]	Abstraction	No difference in abstraction has been found
	HC	40	65.2 (8.9)	50	29.8 (0.5)			
Papp et al. [88]	HC	92	67.4 (8.8)	65.2	29.2 (1.01)	MMSE 24–30 CDR ≤ 0.5	Groton Maze Learning Test	MCI: higher exploratory errors, rule-breaks errors and lower difference in errors between trial 1 and trial 2
	aMCI	59	69.9 (8.1)	45.8	27.7 (1.35)			
Pertl et al. [89]	MCI	22	75	50	27	[1,34]	CLOX–1	MCI: lower planning than HC
	HC	29	73	65.52	29			
Pertl et al. [90]	HC	19	74	-	29	[1,34]	CLOX–1	No difference in planning has been found
	MCI	17	79	70.59	27			
Peters et al. [91]	HC	20	72.0 (6.9)	70	29.6 (0.5)	[31]	TOL	No difference in planning has been found
	MCI	22	70.4 (7.1)	59.1	28.1 (1.4)			
Rainville et al. [92]	HC	42	69.9 (7.3)	-	29.4 (0.9)	[93]	TOL	MCI: higher rule breakings and abandoned than HC
	MCI	51	68.9 (8.3)	-	28.0 (1.6)			
Royall et al. [94]	HC	45	75.8 (6.0)	75.6	27.8 (2.1)	CDR < 3 MMSE < 10	CLOX–1	MCI: lower planning than HC
	MCI	40	78.6 (6.7)	72.5	24.8 (2.9)			
Sánchez–Benavides et al. [95]	HC	30	72.1 (4.7)	51	28.8 (1.2)	[96]	TOL–Drexel Version	No difference in planning has been found
	MCI	23	72.9 (7.4)	61	26.3 (2.1)			
Sánchez–Benavides et al. 2014 [97]	HC	356	64.9 (9.3)	59.6	28.7 (1.5)	[79]	TOL–Drexel Version	MCI: lower total correct than HC MCI: higher total moves, total initiation time, total execution time and total solving time than HC
	MCI	79	72.8 (6.5)	57	25.7 (2.2)			
Sanders et al. [98]	HC	37	70.27 (7.93)	65.57	-	[30,31]	Zoo Map Test	MCI: higher total errors than HC
	MCI	37	72.89 (9.01)	45.94	-			
Schmitter-Edgecombe et al. [99]	MCI	38	70.58 (8.6)	55.26	-	[30,31]	Zoo Map Test	MCI: lower planning than HC
	HC	38	69.34 (7.95)	71.05	-			
Schmitter-Edgecombe et al. [100]	HC	51	70.94 (8.1)	-	-	[30,31]	CLOX–1	MCI: lower planning than HC
	MCI	51	70.98 (8.42)	-	-			
Serra et al. [101]	aMCI	16	72.5 (6.5)	37.5	25.3 (1.2)	[30]	RCPM	No difference in reasoning has been found
	HC	13	64.1 (10.5)	30.77	28.9 (1.3)			
Serra et al. [102]	aMCI	15	70.9 (9.0)	27	25.4 (1.7)	[1,6]	RCPM	No difference in reasoning has been found
	naMCI	13	68.6 (5.7)	77	26.3 (1.6)			
	HC	28	63.4 (8.9)	37	28.4 (1.7)			
Serrao et al. [103]	HC	38	67.37 (5.89)	-	27.88 (0.62)	[1,6]	Matrix Reasoning	MCI: lower IQ than HC
	MCI	61	68.92 (6.49)	-	26.03 (0.44)			
Sheldon et al. [104]	aMCI	16	74.4 (7.4)	69	29.5 (0.7)	[1]	Means-Ends Problem Solving Test	MCI: lower problem solving than MCI
	HC	16	75.1 (5.7)	38	28.4 (1.2)			
Sherod et al. [105]	HC	85	67.2 (8.2)	65	29.4 (0.9)	[30]	CLOX–1 DRS–2 ConceptualizationCognitive Competency Test	CLOX–1 No difference in planning has been found DRS–2 Conceptualization MCI: lower abstraction than HC Cognitive Competency Test MCI: lower abstraction than HC
	MCI	113	70.3 (7.4)	57	28.1 (1.9)			
Tabert et al. [106]	HC	83	66.9 (9.1)	59.4	29.3 (0.8)	[38]	Similarities Mattis Identities and Oddities	Similarities MCI: lower verbal abstract reasoning than HC Mattis Identities and Oddities No difference in nonverbal abstract reasoning has been found
	MCI	148	67.0 (9.9)	55	27.5 (2.2)			
Tam et al. [107]	MCI	24	73.88 (10.8)	50	27.22 (1.65)	MMSE > 24	CLOX–1	No difference in planning has been found
	HC	24	73.25 (9.03)	62.5	28.63 (1.38)			
Tripathi et al. [108]	MCI	22	68.18 (5.7)	27.27	28.0 (2.37)	[1]	TOH	No difference in planning has been found
	HC	20	68.65 (6.0)	25	30.0 (1.0)			
Urbanowitsch et al. [109]	HC	143	73.94 (0.99)	52.45	28.91 (1.12)	[110]	Similarities	MCI: lower reasoning than HC
	MCI	63	74.21 (1.03)	50.79	28.07 (1.41)			
Weakley et al. [111]	MCI	32	69.34 (8.6)	66	-	[1]	Zoo Map Test	No difference in planning has been found
	HC	64	68.13 (9.16)	72	-			
Wu et al. [112]	HC	16	67.75 (5.64)	50	29.13 (1.09)	[1]	Matrix Reasoning Block Design	Matrix Reasoning No difference in IQ has been found Block Design MCI: lower IQ than HC
	aMCI	13	69.0 (5.69)	53.85	26.23 (2.05)			
Zamarian et al. [113]	HC	18	65.1 (4.6)	61.11	29.8 (0.4)	[1]	CLOX–1	MCI: lower planning than HC
	MCI	18	69.0 (7.5)	55.55	26.9 (1.2)			
Zhang et al. [114]	HC	32	73.5 (8.5)	-	28.7 (1.8)	[38]	Trail Making Test (B-A) Porteus Maze Test Verbal Fluency Test (fruits and animals)	Trail Making Test MCI: lower planning than HC Porteus Maze Test MCI: lower planning than HC Verbal Fluency Test MCI: lower planning than HC
	MCI	32	73.7 (8.2)	-	27.4 (2.0)			
Zhang et al. [115]	aMCI	34	67.9 (6.7)	58.82	28.3 (0.5)	[1]	Abstraction–MoCA CDT	Abstraction No difference in abstraction has been found Clock Drawing Test No difference in planning has been found
	HC	36	67.4 (5.0)	50	29.5 (0.7)			
Zheng et al. [116]	aMCI	34	67.9 (6.7)	58.82	28.3 (1.5)	[40]	CDT	No difference in planning has been found
	HC	36	67.4 (5.0)	50	29.5 (0.7)			
Zheng et al. [117]	aMCI	50	69.8 (6.8)	68	27.9 (1.5)	[40]	CDT	No difference in planning has been found
	HC	48	69.2 (5.1)	60.41	29.5 (0.7)			

SD = standard deviations; MMSE = Mini Mental State-Examination; MCI = Mild Cognitive Impairment; aMCI = amnesic Mild Cognitive Impairment; naMCI = non amnesic Mild Cognitive Impairment; MCI+ = Mild Cognitive Impairment multiple domains; aMCI+ = amnesic Mild Cognitive Impairment multiple domain; naMCI+ = non amnesic Mild Cognitive Impairment multiple domains; MCI-na = normal awareness for memory deficits; MCI-pa = poor awareness for memory deficits; RCPM = Raven’s Progressive Coloured Matrices; RPM = Raven’s Progressive Matrices; TOL = Tower of London; TOH = Tower of Hanoi; Tower Test (D-KEFS) = Tower test (Delis–Kaplan Executive Function System); CDT = Clock Drawing Test; WCST-CV = Wisconsin Card Sorting Test–Computer Version; MoCA = Montreal Cognitive Assessment; CDR = Clinical Dementia Rating Scale; NINCDS-ADRDA = National Institute of Neurological and Communicative Disease and Stroke/Alzheimer’s Disease and Related Disorders Associations; AACD = Ageing-Associated Cognitive Decline; DSM = Diagnostic and Statistical Manual of Mental Disorders; CLOX-1 = Clock Drawing Task; DRS-2 =Dementia Rating Scale-2; ACED money management = Assessment of Capacity for Everyday Decision-Making money management; MacCAT-T = The MacArthur Competence Assessment Tool for Treatment.

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
