# Peer review of "Higher-Level Executive Functions in Healthy Elderly and Mild Cognitive Impairment: A Systematic Review"

_jcm, 2022, doi:10.3390/jcm11051204_

Round 1

Reviewer 1 Report

Title: the title is not reflecting the study well as it found a strong prevalence of higher-level executive functions deficits in MCI elderly than in healthy elderly. I suggest to revise it into

Abstract: Please remove the references from the Abstract

Introduction: Please provide references for the sentence “The most studied type is amnesic MCI, in which the subject has a memory disorder that can be at a single domain (aMCI) or multiple domains (aMCI - md).”

Introduction: What is the importance in preforming a systematic review for the relationships between executive Functions and healthy and pathological aging? What are the gaps in the literature?

Study aims: I suggest to make the study aims specific for cognitive ageing as healthy and pathological ageing could be very fast and includes for instance frailty, falls and biomarkers.

Reviewer 2 Report

  1. Since MCI is a heterogenous  disease group, it seems necessary to review at least aMCI and naMCI separately.
  2. Although planning, reasoning, problem-solving, and fluid intelligence are concepts in different categories, they were described at the same time and described confusingly. To talk about fluid intelligence, it seems that some background information is needed, including the concepts of crystalized intelligence and fluid intelligence.
  3. "According to Diamond’s model, the EFs have three major components: inhibition, working memory, attentional control, and cognitive flexibility." In this sentence, you mentioned four components instead of three. This sentence need to be corrected.
  4. You mentioned in the order of planning, reasoning, problem solving, and fluid intelligence in Introduction, and suddenly described in the order of planning, reasoning, fluid intelligence, and problem solving in method and result, but I do not know on what basis the order was changed.
